# MSC and HUVEC co-cultured fillers overcome intractable fistula in a new mouse model

Soichiro Hirasawa[1]*, Kentaro Murakami[1], Masayuki Kano[2], Satoshi Endo[1], Takeshi Toyozumi[1], Yasunori Matsumoto[1], Ryota Otsuka[1], Nobufumi Sekino[1], Tadashi Shiraishi[1], Takahiro Ryuzaki[1], Kazuya Kinoshita[1], Takuma Sasaki[1], Hisahiro Matsubara[1]

1 Department of Frontier Surgery, Graduate School of Medicine, Chiba University, Chiba, Japan,
2 Division of Esophago-Gastrointestinal Surgery, Chiba Cancer Center, Chiba, Japan

☯ These authors contributed equally to this work.
* gifuhirasawa@yahoo.co.jp

## Abstract

Anastomotic leakage can lead to intractable fistulae after gastrointestinal surgery in patients with severe comorbidities. In this study, we aimed to devise new intractable fistula mouse models and evaluate the utility of the fillers containing human mesenchymal stem cells (MSCs) and human umbilical vein endothelial cells (HUVECs). After determining the optimal ratio of MSCs to HUVECs as fillers, we created new intractable fistula mouse models and verified the usefulness of the above-mentioned fillers for these fistulas. As the filler containing a 1:1 ratio of MSC: HUVEC showed the highest expression of FGF2 and VEGF among the organization-forming fillers, we determined that this was the optimal ratio. When this filler was transplanted into irradiated and steroid-treated mice with excisional wounds, the skin defects healed significantly faster in the filler-transplanted group than in the non-transplanted group ($P<0.05$). Furthermore, we established a new mouse model of a gastrointestinal fistula by securing the cecum to the abdominal wall and puncturing the skin, abdominal wall, and intestinal wall with an indwelling needle. The fistula remained patent for at least seven days and was intractable. Unlike the adhesive group (group 1) (0/5) and the group implanted with fillers containing MSCs (group 2) (1/5), all fistulas were closed in the group implanted with fillers containing MSCs and HUVECs (group 3) (5/5). This study demonstrated that a treatment strategy using HUVEC is advantageous for treating intractable fistulae connected to the gastrointestinal tract. HUVEC should be included when fillers are used to close fistulas.

## Introduction

Anastomotic leakage (AL) is one of the most alarming complications of gastrointestinal surgery. Although this incidence has decreased with the advancement of

**Data availability statement:** All relevant data are within the manuscript and its Supporting Information files.

**Funding:** This study was supported in part by a Grant-in-Aid for Scientific Research (KAKENHI: 20K17673) from the Japan Society for the Promotion of Science.

**Competing interests:** The authors have declared that no competing interests exist.

instrumental anastomosis, an analysis of the National Clinical Database (NCD) data, which covers more than 95% of surgical procedures in Japan, shows that anastomotic leakage is observed in 12.6% of esophagectomies [1], 2.3% of distal gastrectomies [2] and 1.8% of right hemicolectomies [3]. It's still not 0%. Advances in drainage technology and nutritional management have made it possible to cure these conditions in many cases; however, some cases are difficult to treat. In recent years, with the advent of an aging society, there have been opportunities to perform surgery in patients with many comorbidities. Among these conditions, severe arteriosclerosis, diabetes, use of steroids, and radiation therapy can delay wound healing and sometimes lead to intractable fistulas that resist conservative treatment [4]. These fistulas can persist for a long time, and managing them with countermeasures can be challenging.

For intractable fistulas, filling the space with materials other than granulation tissue becomes crucial. Although fibrin glue [5], cyanoacrylate [6], and polyglycolic acid mesh [7] have shown promise, concerns remain regarding their safety and long-term efficacy in the body. Additionally, existing options struggle to adequately fill larger spaces due to limited material availability. Therefore, it is necessary to develop a filler that can safely and reliably close fistulas.

Human mesenchymal stem cells (MSCs) are capable of self-renewal and multilineage differentiation into various mesoderm-derived tissues, such as bone, cartilage, fat, and muscle [8,9]. Furthermore, their paracrine effects, via cytokines and extracellular vesicles (EVs) are involved in immunomodulation, inflammation, and tissue regeneration [10]. Because of these capabilities, stem cells have recently played a central role in regenerative medicine in soft tissues [8,9]. In recent years, the use of mesenchymal stem cells for wound healing has increased rapidly, not only in basic research but also in clinical trials. In 2023, the efficacy of adipose tissue-derived mesenchymal stem cells (darvadstrocel) in the treatment of complicated hemorrhoidal fistulas associated with Crohn's disease was reported, with a combined remission rate of 59.1% after 24 weeks and 68.2% after 52 weeks [11]. Thus, the efficacy of MSCs alone in refractory wounds is limited, and a combination of genetic modifications of MSCs and other cell-based therapies is required.

In 2013, Takebe et al. reported that they produced functional liver tissue with vascular structures by co-culturing human umbilical vein endothelial cells (HUVECs) with MSCs [12]. HUVECs, first isolated by Jaffe in 1973, are well known for their ability to secrete vascular endothelial growth factor (VEGF) and promote angiogenesis [13,14]. Intractable fistulas caused by anastomotic leakage are often found to have poor blood flow due to microvessel obstruction. Therefore, promoting angiogenesis at the site of the defect is critical for tissue regeneration. Studies have shown that co-cultures of MSCs and HUVECs can generate tissues with significantly higher vascular density than those derived from MSCs alone [15].

Based on this evidence, we hypothesized that incorporating both MSCs and HUVECs into a cell-based filler could enhance vascularization and improve healing outcomes for gastrointestinal fistulas. While fillers co-cultured with MSCs and HUVECs have been used in adipose and bone tissue engineering [16,17], to our knowledge, their application in gastrointestinal fistula closure has not been reported.

Moreover, to evaluate the therapeutic potential of such fillers in vivo, a reliable and clinically relevant animal model is essential. Although an enterocutaneous fistula model has been previously reported, it resembles a colostomy [18] and does not replicate the complexity of gastrointestinal fistulas. Thus, a novel model that better mimics clinical presentation is needed.

Therefore, the objective of this study was to evaluate whether fillers co-cultured with MSCs and HUVECs could promote angiogenesis and thereby improve the closure rate of intractable gastrointestinal fistulas using a newly established mouse model.

## Materials and methods

### 2.1 Cell culture and creating fillers

Human bone marrow-derived mesenchymal stem cells (hBM-MSCs) were obtained from Lonza (Cat# CC-2501, Basel, Switzerland) and cultured in MSC Growth Medium (MSCGM BulletKit, Cat# PT3001) according to the manufacturer's instructions. Cells were used between passages 3 and 5. The MSC identity was confirmed by the supplier using established surface marker profiles, including CD73, CD90, CD29, CD105, CD166, and CD44 positivity, and the absence of CD14, CD19, CD34, and CD45, in accordance with ISCT criteria.

Human umbilical vein endothelial cells (HUVECs) were obtained from Lonza (Cat# CC-2517, Basel, Switzerland) and cultured in EGM-2 medium (Lonza, Cat# CC-3162). Cells were used between passages 3 and 6. According to the supplier, HUVECs were characterized based on endothelial marker expression and angiogenic capacity.

Six hundred microliters (600 µL) of Matrigel Matrix growth factor-reduced (Corning, Arizona, USA) were used to coat each well of a 12-well plate, followed by a 60-minute incubation at 37°C. Cocultures of MSCs and HUVECs were initiated at five different ratios (0:1, 1:3, 1:1, 3:1, and 1:0) to achieve a total cell count of $1.6 \times 10^7$. The co-culture ratios of MSCs to HUVECs were selected based on previously reported studies in which these cell types were combined to fabricate tissue fillers [19]. MSC and HUVEC were maintained in an endothelial cell growth medium. The filler was created by culturing human MSCs and HUVECs on Matrigel under three-dimensional (3D) conditions. This approach resulted in a tissue-like structure, with Matrigel serving as a biological scaffold to support the formation of a compact cellular mixture. No additional biopolymers, such as hyaluronic acid, were used.

### 2.2 Quantitative reverse-transcription PCR

Co-cultures of MSCs and HUVECs were initiated at three different ratios (MSCs:HUVECs = 1:1, 3:1, and 1:0), with a total of $1.6 \times 10^7$ cells per sample. The resulting tissues were homogenized by pipetting, and total RNA was extracted using QIAzol Lysis Reagent (Qiagen, Venlo, Netherlands), following the manufacturer's instructions. Complementary DNA (cDNA) was synthesized using the High-Capacity RNA-to-cDNA Kit (Thermo Fisher Scientific, Waltham, MA, USA). Quantitative RT-PCR was performed in triplicate using a MyiQ™2 Two-Color Real-Time PCR Detection System (Bio-Rad, Hercules, CA, USA) and TB Green Fast qPCR Mix (TaKaRa, Shiga, Japan). Primers for fibroblast growth factor 2 (FGF2) and vascular endothelial growth factor (VEGF) were designed using Primer3Plus (http://www.bioinformatics.nl/cgi-bin/primer3plus/primer3plus.cgi/) and synthesized by Eurofins Genomics (Tokyo, Japan). Detailed primer sequences, annealing temperatures, and product sizes are provided in **Supporting Information Table1. qPCR analyses were independently performed in triplicate (n = 3).

### 2.3 Immunohistochemistry

Tissues with MSCs and HUVECs in a ratio of 1:1 were fixed in 4% paraformaldehyde for 1 d. The sections were then embedded in paraffin. The cells were permeabilized and blocked for 30 min. CD31 was used as a specific marker for HUVECs and CD44 for MSCs. Primary mouse anti-human CD31 (ab218, Abcam plc, Cambridge, UK, dilution 1:100) or

rabbit anti-human CD44 (ab51037, Abcam plc, Cambridge, UK, dilution 1:100) antibodies (S0809, Agilent, CA, USA) were added to the tissue and incubated at 37°C for 60 min. A secondary goat anti-rabbit IgG, Alexa Fluor 488 (A11008, Thermo Fisher Scientific, Massachusetts, United States, 4 $\mu$g/mL) and goat anti-mouse IgG, Alexa Fluor 594 (A11005 Thermo Fisher Scientific, Massachusetts, United States, 5 $\mu$g/mL) were applied in antibody diluent and incubated in the dark at room temperature for 60 min. The cell nuclei were counterstained by DAPI (P36935, Thermo Fisher Scientific, Massachusetts, United States, 1 $\mu$g/mL) solution for five minutes. The fluorescent staining was recorded by a Fluorescence microscope (BZ-X700, Keyence, Osaka, Japan).

## 2.4 Transplantation of filler to subcutis

Male BALB/cAJcl-nu/nu mice (ages 6–8 ages weeks old) were used in this study. The mice were anesthetized using a triple anesthetic (medetomidine (Nippon Zenyaku Kogyo, Hukushima, Japan), butorphanol tartrate (Meiji Seika, Tokyo, Japan), and midazolam (Sandoz, Tokyo, Japan)). They were transplanted with a MSCs filler: HUVECs at a 1:1 or 1:0 ratio. On day seven, the mice were sacrificed by cervical dislocation, and the skin of the transplanted area was resected. The tissues were fixed in paraformaldehyde and stained with hematoxylin. Capillary densities of the fillers were determined using a microscope (BZ-9000, Keyence, Osaka, Japan). Capillary density analysis in the subcutaneous filler transplantation model was performed in triplicate (n = 3).

## 2.5 Creation of mouse model with intractable enterocutaneous fistula

Male BALB/cAJcl-nu/nu mice (ages 8−10 weeks old) were anesthetized (n = 5). The abdomen, abdominal wall, and cecum were fixed using two stitches of 6−0 nylon (Bear Medic, (Ibaraki, Japan)). The skin, abdominal wall, and intestines were punctured using a 10-gauge indwelling needle (BD Angiocath, BD, New Jersey, USA). Only the outer tube was implanted. Three days later, the outer tube was removed creating an enterocutaneous fistula. In response to weight loss, the mice were supplemented with saline (Otsuka Pharmaceutical Factory, Tokushima, Japan). Over seven days, the enterocutaneous fistula was monitored. On the seventh day, the area was sampled and stained with hematoxylin and eosin (H&E).

## 2.6 Transplantation of filler to mouse model with intractable enterocutaneous fistula

An enterocutaneous fistula mouse model was established using the method above. Three days after removing the outer tube, the mice were divided into three groups: an adhesive group coated with only physiological tissue adhesive (Group1: n = 5), a transplanted group with fillers containing MSCs (Group2: n = 5), and a transplanted group with fillers containing HUVECs and MSCs in a ratio of 1:1 (Group3: n = 5). Mice in the adhesive group received a coating of physiological tissue adhesive directly onto their fistulae. The filler-transplanted group, on the other hand, had the fillers applied to their fistulae, followed by application of a physiological tissue adhesive to secure the filler and fix it in place. Both groups had their fistulae monitored for seven days, and in the filler-transplanted group, the area around the filler was then sampled and stained with (H&E). For the in vivo intractable fistula model, **five mice were used per group (n = 5)**. Due to the substantial cost and technical difficulty associated with both the creation of the fistula model and the preparation of the cell-based fillers, **a larger sample size was not feasible**.

## 2.7 Animals

This study was approved by the Chiba University Animal Experiment Committee (approval number: 4–157), and all procedures were conducted in accordance with the Chiba University Institutional Guidelines for the Use of Laboratory Animals. Mice were housed in individually ventilated cages under specific pathogen-free (SPF) conditions, with a 12-hour light/dark cycle, ambient temperature maintained at 22 ± 2 °C, and relative humidity at 55 ± 10%. Food and water were available ad libitum. Anesthesia was induced using a triple-anesthetic combination consisting of medetomidine, butorphanol tartrate,

and midazolam. Animals were monitored postoperatively to minimize suffering. Humane endpoints were defined, and euthanasia was performed by cervical dislocation under deep anesthesia, following institutional guidelines.

### 2.8 Statistical analysis

Data were analyzed using a Student's t-test conducted using Excel software (Microsoft, Redmond, WA, USA). Statistical significance was set at $P < 0.05$.

## Results

### 3.1 Preparation of MSC-HUVEC fillers

Co-cultures of MSCs and HUVECs were initiated in five different ratios (MSCs: HUVECs 0:1, 1:3, 1:1, 3:1, and 1:0) to achieve a total cell count of $1.6 \times 10^7$ (Fig 1a). Co-cultures of MSCs and HUVECs at three ratios (MSCs: HUVECs 1:1, 3:1, and 1:0) formed organized structures.

### 3.2 Quantification of angiogenesis markers

Tissues with MSCs: HUVECs ratios of 1:1, 1:3, and 1:0 were homogenized and subjected to quantitative reverse-transcription PCR to examine the expression of marker genes involved in angiogenesis (Fig 1b). Tissues with an MSCs: HUVECs ratio of 1:1 showed the highest expression of FGF2 and VEGF. The full qPCR dataset, including Ct values, ΔCt/ΔΔCt calculations, and expression ratios for each sample and group, are available in S1(FGF2), S2(VEGF) Data. Based on these results, we determined that a 1:1 ratio was optimal for the filler.

### 3.3 Immunostaining of MSCs and HUVECs

Skin tissues co-cultured with MSCs and HUVECs at a 1:1 ratio were stained with fluorescent antibodies specific for MSCs (green) and HUVECs (red) and visualized using immunofluorescence microscopy (Fig 1c). The results showed that MSCs and HUVECs were mixed to form tissues. Immunostaining was used to observe the spatial distribution and interaction of MSCs and HUVECs within the filler, but not for determining the optimal co-culture ratio.

### 3.4 Transplantation of filler to subcutis

Next, the changes in the prepared filler, as shown in Fig 1, were observed in vivo. H&E staining of the filler with MSCs and HUVECs at a ratio of 1:1 administered to mouse subcutaneous tissue showed that the filler had grown under the skin (Fig 2a). To evaluate the function of HUVECs in angiogenesis, capillary density was determined (Fig 2b). The results showed that the capillary density in a filler with MSCs and HUVECs at a ratio of 1:1 was $422.7/mm^2$, which was significantly higher than that in a filler at a ratio of 1:0 ($53.9/mm^2$). Raw data and summary statistics for these comparisons are available in Supporting Information S3 Data.

### 3.5 Mouse model with intractable enterocutaneous fistula

To create a fistula mouse model, the cecum was secured to the abdominal wall, and an indwelling needle was inserted through the skin, abdominal wall, and intestinal wall. Only the outer tube of the indwelling needle remained, creating a permanent opening (fistula) (Fig 3). Three days later, fistula formation was confirmed after removal of the outer tube. Seven days after removal of the indwelling needle, all caecal fistulas (n = 5) remained open. Hematoxylin and eosin staining confirmed fistulous tract formation from the intestine to the subcutis.

### 3.6 Transplantation of fillers in a mouse model with intractable enterocutaneous fistula

To evaluate the usefulness of filler implants in enterocutaneous fistulae, the healing process was compared among an adhesive group (Group1), a transplanted group with fillers containing MSCs (Group2), and a transplanted group with fillers

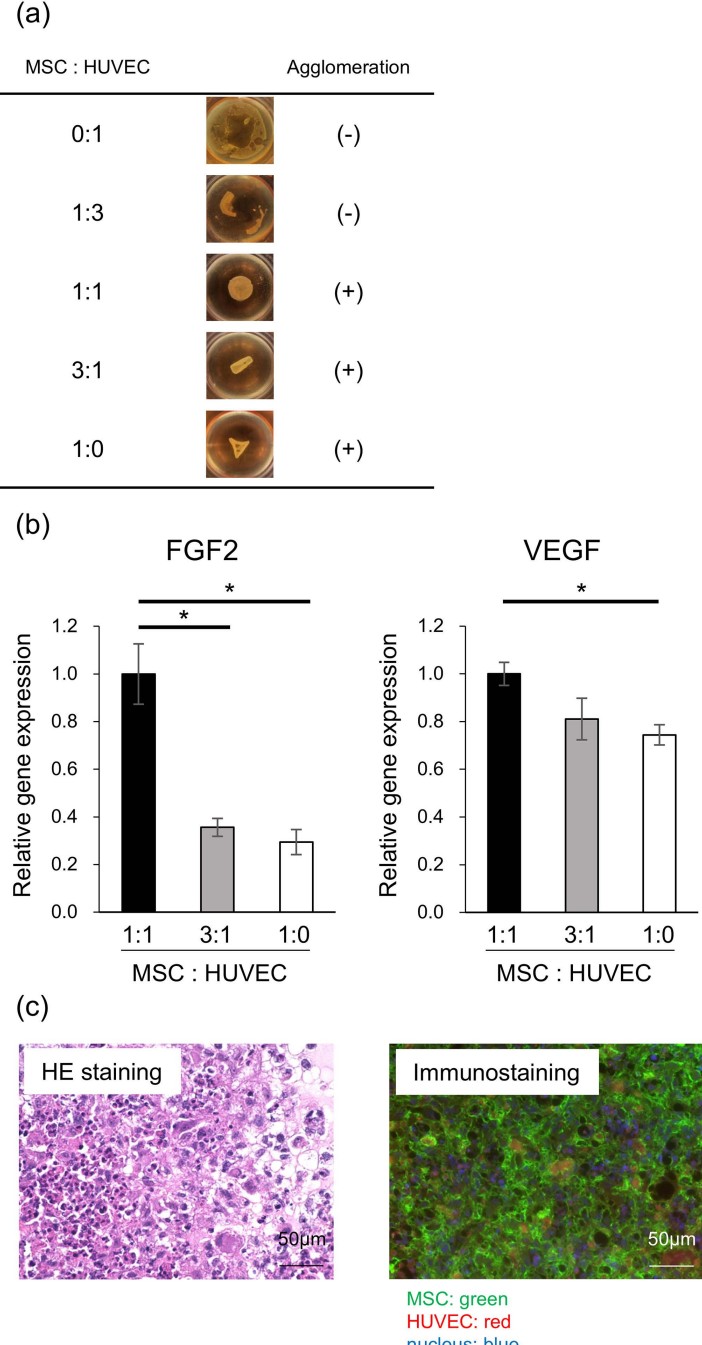

**(a)**

| MSC : HUVEC | | Agglomeration |
|---|---|---|
| 0:1 | | (-) |
| 1:3 | | (-) |
| 1:1 | | (+) |
| 3:1 | | (+) |
| 1:0 | | (+) |

**(b)**

FGF2 / VEGF — Relative gene expression vs MSC : HUVEC (1:1, 3:1, 1:0)

**(c)**

HE staining — 50µm

Immunostaining — 50µm

MSC: green
HUVEC: red
nucleus: blue

**Fig 1. Preparation of MSC and HUVEC filler.** (a) Co-cultures of MSCs and HUVECs in five ratios. (b) mRNA expression of FGF2 and VEGF in the co-cultured tissues. (c) HE and immunostaining of the co-cultured tissues (MSC: Green, HUVEC: Red). Scale bar=50µm. The error bars represent mean ± standard deviation. These data were analyzed for statistical significance using Student's t-test: *$P<0.05$. Abbreviations: MSC, mesenchymal stem cell; HUVEC, human umbilical vein endothelial cell; FGF2, fibroblast growth factor2; VEGF, vascular endothelial growth factor.

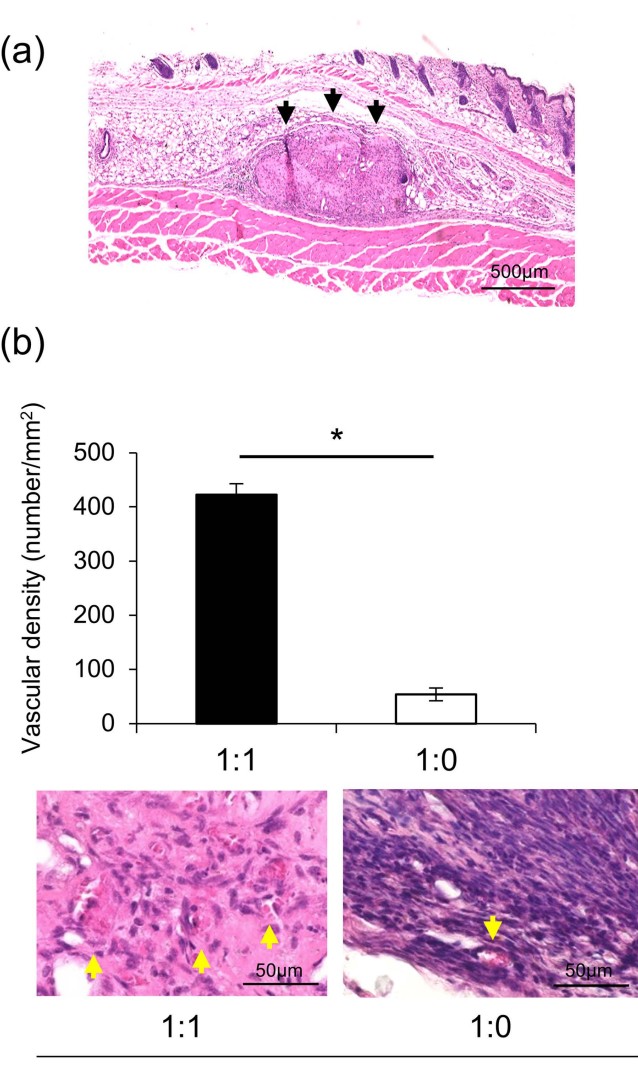

**Fig 2. The filler transplanted into the mouse subcutaneous space.** (a) HE staining of the filler transplanted into the mouse subcutaneous space (granulation formed by the filler: black arrows). (b) Comparison of vascular density of the transplanted fillers mixed with different MSC and HUVEC proportions (vascular: yellow arrows). The error bars represent mean ± standard deviation. These data were analyzed for statistical significance using Student's t-test: *$P<0.05$. Abbreviations: MSC, mesenchymal stem cell; HUVEC.

containing HUVECs and MSCs (Group3) (Fig 4). In group 1, none of the 5 mice(0/5) achieved fistula closure and continued to experience leakage of intestinal fluid. In group 2, one out of 5 mice (1/5) successfully closed their fistula.

In group 3, all 5 mice (5/5) achieved fistula closure via granulation tissue formation by day 3. Their wounds were completely covered with skin by day seven. H&E staining showed that in group 3, granulation had formed in the area of the fistula, and the injured part of the intestine was closed. Vessel-like structures are observed within the granulation tissue.

## Discussion

In this study, we created a new intractable fistula model and evaluated the usefulness of fillers co-cultured with MSCs and HUVECs. Tissues co-cultured with MSCs and HUVECs at a 1:1 ratio showed the highest expression of angiogenic

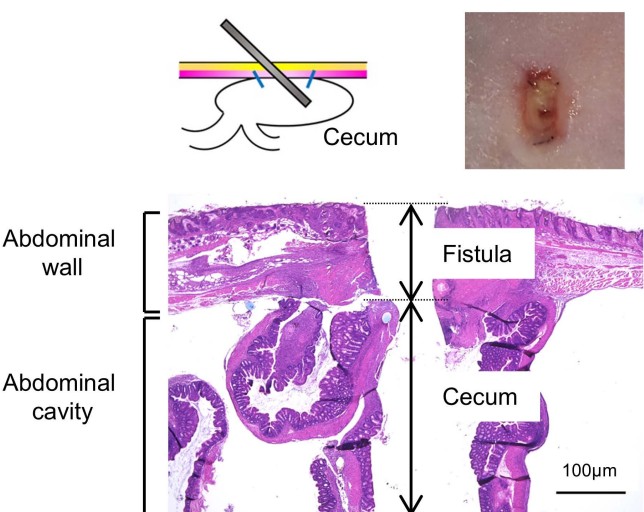

**Fig 3. Creation of a mouse model of intractable enterocutaneous fistula.** Gross and microscopic images of the enterocutaneous fistula were shown.

markers and the highest blood vessel density within the tissue; therefore, we decided to use this tissue as a filler. We established a novel mouse fistula model by securing the cecum to the abdominal wall. This model resulted in an intractable fistula that resisted spontaneous healing. Notably, the filler described above demonstrated a high success rate in achieving closure of these fistulas.

To assess the potential of fillers in treating refractory gastrointestinal fistulas (occurring after cancer surgery), we developed a reliable and easily observable mouse model replicating key features of these clinically challenging non-healing fistulas. Although a model similar to a prosthesis has been previously reported, it differs from a fistula that develops after suture failure, which was our target in this study, because the mucosa is directly continuous with the skin [13]. Because there have been no adequate reports to date, we devised a new fistula model at the beginning of this study. First, a fistula connecting the gastrointestinal tract to the skin was created for ease of observation. However, fistulas in the intestinal tract often result in death due to the massive leakage of intestinal fluid. Therefore, we focused on the mouse cecum. Because the mouse cecum is long and does not produce much stool, the general condition of the mice with a fistula connecting the cecum to the skin was stable, and the fistula did not close spontaneously. We speculate that this model will be useful for future research on fistulas.

Recently, tissue regenerative medicine for enterocutaneous fistulae using MSC has been expected, such as MSC sheets, [9] and sutures made from MSC [10]. However, these all have a small amount of tissue, and cannot close fistulae with large spaces. A major cause of failure in transplantation of regenerated tissue is inadequate vascularization of the tissue [11]. Enterocutaneous fistulae are often associated with poor blood flow; thus the tissue needs to be vascularized and viable. Similarly, in this study, the complete healing of refractory fistulas created in mice using stem cells alone was difficult to achieve.

HUVEC, isolated from the umbilical vein by Jaffe in 1973, secretes VEGF and promotes angiogenesis. Several studies have reported good results using fillers with HUVECs alone or HUVEC-derived exosomes [20], but the method of making fillers has not been established, as it differs in each study. In this study, VEGF and FGF2 were selected as representative markers of angiogenesis, which is a critical factor for the engraftment and functional integration of the filler. Both genes are well-established proangiogenic factors that directly contribute to vascular network formation. Moreover, these genes have frequently been used in previous studies evaluating MSC and HUVEC co-cultures, especially when assessing

| | Group1 (n=5) | Group2 (n=5) | Group3 (n=5) |
|---|---|---|---|
| adhesive | + | + | + |
| MSC | - | + | + |
| HUVEC | - | - | + |
| Day 0 | | | |
| Day 3 | | | |
| Day 7 | | | |
| number of successful closure | 0/5 | 1/5 | 5/5 |

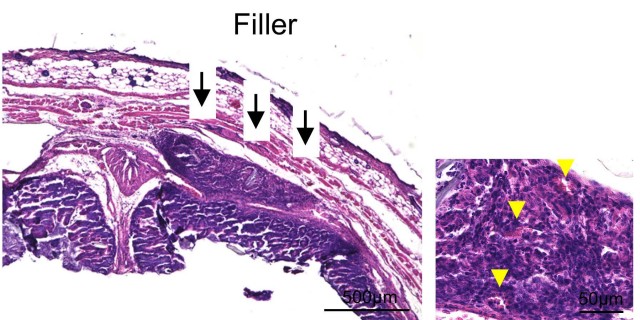

**Fig 4. Treatment results with filler in each group.** Microscopic images of the granulation formed by the filler were shown (granulation formed by the filler: black arrows, vascular: yellow arrows). Abbreviations: MSC, mesenchymal stem cell; HUVEC.

optimal cell ratios for tissue integration and angiogenic potential [21,22]. Based on this rationale and our own observations, we considered VEGF and FGF2 appropriate indicators of tissue organization capacity in our filler model. When MSCs and HUVECs were mixed in a 1:1 ratio, the tissue with the highest VEGF expression was formed. Previous reports have shown that tissues with a 1:1 or 1:3 ratio of MSCs to HUVECs have the highest expression of angiogenic markers, and the results of this study are consistent with those of previous reports [19,21]. Furthermore, filler-derived granulation

tissue prepared at this ratio had a significantly higher vascular density and histopathological angiogenesis than granulation tissue without HUVECs. Direct or indirect interactions between MSCs and HUVECs, as well as VEGF secreted by HUVECs, have been suggested as reasons why combination therapy with these cells may favor wound healing. Beloglazova et al. reported that MSCs support the formation of EC tubular networks (ETNs) via the urokinase-type plasminogen activator (uPA) system when co-cultured with ECs [22]; Chance et al. reported that adipose-derived EVs promote HUVEC tube formation, and fat-derived EVs promote HUVEC tube formation [23].

Based on these results, we implanted fillers in our newly created mouse model of intractable fistulas and found that fillers with MSCs alone failed to achieve complete healing. In contrast, fillers co-cultured with HUVECs and MSCs achieved healing in all cases. The relatively low efficacy of MSC-only fillers observed in this study may be due to the challenging microenvironment of the gastrointestinal fistula model, which simulates clinical enterocutaneous fistulas. In such conditions, MSCs alone may not sufficiently promote tissue integration. However, co-administration of HUVECs likely enhanced angiogenesis, thereby supporting better engraftment of the filler. To the best of our knowledge, this is the first time that a filler co-cultured with MSCs and HUVECs is effective in closing refractory fistulas involving the gastrointestinal tract. Recently, the function of this filler mixture in the repair of myocardial and cerebral infarctions has been analyzed [24,25] and it is expected to be useful in many areas of regenerative medicine. In the future, this mixed filler is expected to benefit patients undergoing gastrointestinal surgery.

However, this study has two limitations. First, using immunosuppressed mice meant that the assessment of potential rejection, an important factor in clinical applications, was not possible. While MSCs generally have low immunogenicity [26], evaluating the immunogenicity of HUVECs in this context is crucial. Utilizing HUVECs derived from autologous induced pluripotent stem cells could address potential immune-related concerns. Second, we were unable to identify the mechanism of interaction between MSCs and HUVECs. Once this is clarified, it may be possible to develop the filler to a higher standard.

## Conclusions

In conclusion, using a realistic and observable model, we demonstrated the superiority of fillers co-cultured with MSCs and HUVECs. As this has not yet been verified in a realistic clinical model, we report that using HUVECs is advantageous for treating gastrointestinal fistulas. Therefore, HUVECs should be added to MSCs when fillers are used to close refractory gastrointestinal fistulas.

## Supporting information

**S1. Data qPCR FGF2.**
(XLSX)

**S2. Data qPCR VEGF.**
(XLSX)

**S3. Data.**
(XLSX)

## Acknowledgments

We would like to thank Keiko Iida for their help with this research.

## Author contributions

**Conceptualization:** Soichiro Hirasawa, Masayuki Kano, Satoshi Endo, Takeshi Toyozumi, Yasunori Matsumoto, Ryota Otsuka, Nobufumi Sekino, Takahiro Ryuzaki, Kazuya Kinoshita, Takuma Sasaki.

**Data curation:** Soichiro Hirasawa, Satoshi Endo, Ryota Otsuka, Nobufumi Sekino, Takahiro Ryuzaki, Kazuya Kinoshita, Takuma Sasaki.

**Formal analysis:** Soichiro Hirasawa, Satoshi Endo, Nobufumi Sekino, Takahiro Ryuzaki, Takuma Sasaki.

**Funding acquisition:** Soichiro Hirasawa, Kentaro Murakami.

**Investigation:** Soichiro Hirasawa, Tadashi Shiraishi.

**Methodology:** Soichiro Hirasawa, Masayuki Kano, Takeshi Toyozumi, Yasunori Matsumoto, Tadashi Shiraishi.

**Project administration:** Soichiro Hirasawa, Tadashi Shiraishi.

**Resources:** Soichiro Hirasawa.

**Software:** Soichiro Hirasawa, Kazuya Kinoshita.

**Supervision:** Soichiro Hirasawa, Hisahiro Matsubara.

**Validation:** Soichiro Hirasawa.

**Visualization:** Soichiro Hirasawa, Kentaro Murakami.

**Writing – original draft:** Soichiro Hirasawa.

**Writing – review & editing:** Soichiro Hirasawa.

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
