## [Decision Letter · Decision Letter 0]

27 May 2025

plosone@plos.org

We look forward to receiving your revised manuscript.

Kind regards,

Keykavoos Gholami

Academic Editor

PLOS ONE

Journal Requirements:

This study was supported in part by a Grant-in-Aid for Scientific Research (KAKENHI: 20K17673) from the Japan Society for the Promotion of Science.

5. We note that your Data Availability Statement is currently as follows: All relevant data are within the manuscript and its Supporting Information files.

6. Your abstract cannot contain citations. Please only include citations in the body text of the manuscript, and ensure that they remain in ascending numerical order on first mention.

Reviewers' comments:

Reviewer's Responses to Questions

**Comments to the Author**

1. Is the manuscript technically sound, and do the data support the conclusions?

Reviewer #1: Yes

Reviewer #2: Partly

2. Has the statistical analysis been performed appropriately and rigorously?

Reviewer #1: Yes

Reviewer #2: Yes

3. Have the authors made all data underlying the findings in their manuscript fully available?

Reviewer #1: No

Reviewer #2: Yes

4. Is the manuscript presented in an intelligible fashion and written in standard English?

Reviewer #1: Yes

Reviewer #2: No

Reviewer #1: I have reviewed the manuscript entitled “MSC and HUVEC co-cultured fillers overcome intractable fistula in a new mouse model.” The study addresses an important clinical issue, namely the treatment of intractable fistulae after gastrointestinal surgery, by introducing a novel mouse model and evaluating the therapeutic potential of fillers containing human MSCs and HUVECs. The findings are promising and could have meaningful translational implications. However, several methodological and reporting concerns should be addressed before the manuscript can be considered for publication.

Methodological Concerns

Cell Identity and Characterization:

The manuscript does not provide sufficient detail on the identity and source of the cells used (MSC and HUVEC). For reproducibility and validation, please include information on how these cells were characterized (e.g., surface marker profiles, passage numbers).

Provide catalog numbers or supplier information for both HUVECs and MSCs to enhance transparency.

Primer Information:

The manuscript lists primer sequences within the text. For clarity and ease of reference, I strongly recommend presenting all primer sequences and related details (gene, sequence, annealing temperature, product size) in a table.

Animal Care and Maintenance:

The section describing animal maintenance and welfare procedures is brief. Please expand on husbandry conditions, ethical approval information, and any measures taken to minimize animal suffering.

Results and Data Interpretation

Selection of MSC:HUVEC Ratios:

The rationale for selecting the specific co-culture ratios (1:1, 1:3, and 1:0) of MSCs to HUVECs is unclear. Please elaborate on why these ratios were chosen and whether preliminary experiments or relevant literature informed these choices.

Choice of Gene Expression Analysis:

Only VEGF and FGF2 were analyzed as markers for organization-forming ability. Could the authors clarify why these two genes were chosen? It would strengthen the manuscript to discuss whether other angiogenic or tissue remodeling factors were considered, and provide justification for focusing on these particular genes.

Reviewer #2: Dear Editor,

I am grateful to be considered as a reviewer for the paper entitled "MSC and HUVEC co-cultured fillers overcome intractable fistula in a new mouse model" submitted to PLOS ONE.

This manuscript holds significant translational potential and addresses an essential issue in gastrointestinal surgeries known as intractable fistulae. Although the authors have presented the topic clearly and developed an innovative model with encouraging findings, I recommend addressing several issues to enhance the overall quality of the paper.

1- The introduction fails to explain the impact and necessity of MSCs in this context. What makes them important for this therapeutic filler, and in what ways do they support HUVECs in healing fistulas?

2- The introduction lacks a clear statement of the study’s final objective. Is it focused on the model’s success rate, fistula closure, inflammation control, angiogenesis promotion, or pain and discharge inhibition?

3- What is the origin of the MSCs employed in this research? Are they xenogeneic, allogeneic, or autologous? Additionally, do they originate from adipose tissue, bone marrow, or another source?

4- Could you clarify the composition of the filler? Does it contain a biopolymer such as hyaluronic acid, or is it simply a mixture of cells? This should be clearly specified in the materials and methods part of the manuscript.

5- On page 7, line 3, details about the microscope need to be added.

6- How many times were the experiments repeated? The statistical analysis part does not mention if the experiments were performed in triplicate.

7- The results do not specify how immunostaining was used to determine that a 1:1 ratio of MSC to HUVEC is optimal.

8- This study would benefit from including an additional experimental group with filler containing only HUVECs to assess their impact independent of MSCs. The inclusion of this data would improve the overall understanding of the findings.

9- The authors need to explain why the efficacy of filler containing only MSCs is significantly lower compared to previous studies.

10- This paper would benefit from English language improvement and a more coherent content organization.

**Do you want your identity to be public for this peer review?** For information about this choice, including consent withdrawal, please see our Privacy Policy

Reviewer #1: **Yes: ** Iman Menbari Oskouie

Reviewer #2: No

---

## [Author Response · Author response to Decision Letter 1]

23 Jun 2025

Dear Editor and Reviewers,

Thank you very much for your thoughtful and constructive comments on our manuscript entitled "MSC and HUVEC co-cultured fillers overcome intractable fistula in a new mouse model." We appreciate the opportunity to revise and improve our manuscript. Below, we provide point-by-point responses to each of the comments raised by the reviewers and the editorial office. All changes in the revised manuscript are clearly marked for your convenience.

Reviewer #1:

1. Cell Identity and Characterization:

Comment: The manuscript does not provide sufficient detail on the identity and source of the cells used (MSC and HUVEC). For reproducibility and validation, please include information on how these cells were characterized (e.g., surface marker profiles, passage numbers).

Response: We appreciate the reviewer’s comment regarding the need for more detailed information about the source and characterization of the cells used in our study. In response, we have revised the Methods section to include the supplier, catalog numbers, passage numbers, and cell characterization details for both MSCs and HUVECs.

Specifically, human mesenchymal stem cells (MSCs) were purchased from Lonza (Cat# CC-2501, Basel, Switzerland) and cultured using MSCGM BulletKit (Cat# PT3001). Cells were used between passages 3 and 5. According to the manufacturer, MSC identity was confirmed using standard surface marker profiles consistent with ISCT criteria, including positivity for CD73, CD90, CD29, CD105, CD166, and CD44, and negativity for CD14, CD19, CD34, and CD45.

Human umbilical vein endothelial cells (HUVECs) were also obtained from Lonza (Cat# CC-2517, Basel, Switzerland) and cultured in EGM-2 medium (Lonza, Cat# CC-3162). Cells were used between passages 3 and 6. As per the supplier’s certification, HUVECs were characterized based on endothelial marker expression and angiogenic functionality.

This revised information can now be found in the Methods section (page 5, lines 10–20).

2. Primer Information:

Comment: The manuscript lists primer sequences within the text. For clarity and ease of reference, I strongly recommend presenting all primer sequences and related details (gene, sequence, annealing temperature, product size) in a table.

Response: We have added a new table (Table1) listing all primers used, including gene names, sequences, annealing temperatures, and product sizes.

3. Animal Care and Maintenance:

Comment: The section describing animal maintenance and welfare procedures is brief. Please expand on husbandry conditions, ethical approval information, and any measures taken to minimize animal suffering.

Response:

we have revised the Materials and Methods section to provide more detailed information on animal husbandry conditions, ethical approval, and procedures used to minimize animal suffering.

Specifically, we now include the approval number from the Chiba University Animal Experiment Committee, and we describe the housing environment (SPF conditions, temperature, humidity, light cycle), the anesthetic regimen (medetomidine, butorphanol tartrate, and midazolam), and humane endpoints. The revised description can be found in the Materials and Methods section (page 9, lines 6–15).

4.Selection of MSC:HUVEC Ratios:

Comment: The rationale for selecting the specific co-culture ratios (1:1, 1:3, and 1:0) of MSCs to HUVECs is unclear. Please elaborate on why these ratios were chosen and whether preliminary experiments or relevant literature informed these choices.

Response: The co-culture ratios of MSCs to HUVECs were selected based on previously reported studies in which these cell types were combined to fabricate tissue fillers.

In our own preliminary experiments described in Section 2.1, co-cultures at three specific ratios (MSCs:HUVECs = 1:1, 3:1, and 1:0) successfully formed organized tissue-like structures.

Therefore, we adopted the same ratios for subsequent analyses, including qPCR.

This rationale has now been clarified in the Materials and Methods section (page 6, lines2 –4, page7 , lines 11–12).

5. Choice of Gene Expression Analysis:

Comment: Only VEGF and FGF2 were analyzed as markers for organization-forming ability. Could the authors clarify why these two genes were chosen? It would strengthen the manuscript to discuss whether other angiogenic or tissue remodeling factors were considered, and provide justification for focusing on these particular genes.

Response: In this study, we focused on VEGF and FGF2 because they are well-established key regulators of angiogenesis and are directly involved in vascular network formation and tissue integration. Since successful engraftment of the filler depends heavily on the promotion of angiogenesis, we selected these two representative proangiogenic factors for gene expression analysis.

Furthermore, several previous studies that examined the optimal MSC:HUVEC co-culture ratios for tissue engineering applications have also used VEGF and FGF2 as primary markers of angiogenic potential. Therefore, we considered the selection of these genes to be a valid and relevant approach for evaluating the organization-forming ability of our co-culture system.

We have added this rationale to the Discussion section (page 13, lines 16–23).

Reviewer #2:

1. Role of MSCs in Filler:

Comment: 1- The introduction fails to explain the impact and necessity of MSCs in this context. What makes them important for this therapeutic filler, and in what ways do they support HUVECs in healing fistulas.

Response: We have revised the Introduction to clarify the role of mesenchymal stem cells (MSCs) in our therapeutic approach. While MSC alone possess regenerative and immunomodulatory properties through paracrine signaling, several studies have demonstrated that co-culture with HUVEC significantly enhances angiogenesis compared to MSCs alone. For example, Takebe et al. (2013) showed that MSC–HUVEC co-cultures could form functional vasculature in engineered tissues. Similarly, Jinling Ma et al. (2014) reported that tissues formed from MSCs and HUVECs exhibited significantly higher blood vessel density than those formed from MSCs alone.

2. Clarify Study Objective:

Comment: The introduction lacks a clear statement of the study’s final objective. Is it focused on the model’s success rate, fistula closure, inflammation control, angiogenesis promotion, or pain and discharge inhibition?

Response: We agree that the study objective needed clarification. We have revised the final paragraph of the Introduction to clearly state the aim of the study. The primary objective of this work is to improve the fistula closure rate using a cell-based filler composed of MSCs and HUVECs, leveraging their angiogenic potential to enhance tissue regeneration. This statement now appears at the end of the Introduction section (page 5, lines 5–7).

3. Source of MSCs:

Comment: What is the origin of the MSCs employed in this research? Are they xenogeneic, allogeneic, or autologous? Additionally, do they originate from adipose tissue, bone marrow, or another source?

Response: The mesenchymal stem cells (MSCs) used in this study were human bone marrow-derived MSCs, purchased from Lonza (Cat# CC-2501, Basel, Switzerland). These cells are xenogeneic relative to the recipient mice. We have clarified this information in the Materials and Methods section (page 5, lines 11–16).

4. Composition of Filler:

Comment: Could you clarify the composition of the filler? Does it contain a biopolymer such as hyaluronic acid, or is it simply a mixture of cells? This should be clearly specified in the materials and methods part of the manuscript.

Response: We have clarified the composition of the filler in the Materials and Methods section. The filler used in this study consisted of a three-dimensional co-culture of human MSCs and HUVECs on Matrigel, which served as a biological scaffold. No other biopolymers, such as hyaluronic acid, were included in the formulation.(page 6, lines 4–8).

5. Microscope Details:

Comment: On page 7, line 3, details about the microscope need to be added.

Response: We have added the specific model and manufacturer of the microscope used (BZ-X9000, Keyence, Osaka, Japan) to the Methods section (Page 8, Line 1).

6. Experiment Repetition and Statistics:

Comment: How many times were the experiments repeated? The statistical analysis part does not mention if the experiments were performed in triplicate.

Response: We have clarified the number of replicates in the Materials and Methods section. Specifically, qPCR experiments and capillary density analysis in the subcutaneous filler transplantation model were performed in triplicate (n = 3).

For the in vivo intractable enterocutaneous fistula model, each group included five mice (n = 5). Due to the substantial cost and technical difficulty involved in both creating the fistula model and preparing the cell-based fillers, further expansion of the sample size was not feasible. We have added this information to Methods section (Page 6, Line 22-23, Page 8 Line 2, Page9, Line1-4).

7. Immunostaining and Ratio Determination:

Comment: The results do not specify how immunostaining was used to determine that a 1:1 ratio of MSC to HUVEC is optimal.

Response: We apologize for the confusion. Immunostaining in this study was not used to determine the optimal MSC:HUVEC ratio. Rather, it was performed to visualize how MSCs and HUVECs mixed and organized within the filler structure. The 1:1 ratio was identified as optimal based on qPCR results (FGF2 and VEGF expression) and capillary density measurements, not by immunostaining. We have clarified this distinction in the revised Results sections.

8. Missing HUVEC-only Group:

Comment: This study would benefit from including an additional experimental group with filler containing only HUVECs to assess their impact independent of MSCs. The inclusion of this data would improve the overall understanding of the findings. Recommend including a group with HUVEC-only filler.

Response: We agree that evaluating the independent contribution of HUVECs could provide valuable insight. However, in our preliminary experiments, we found that fillers composed only of HUVECs were unable to form organized tissue structures when cultured on Matrigel. Unlike MSCs, HUVECs alone did not maintain sufficient viability or structure under the same 3D culture conditions. Therefore, it was not feasible to include a HUVEC-only group in the main experiments, as no viable tissue could be produced.

9. MSC-only Efficacy Compared to Literature:

Comment: The authors need to explain why the efficacy of filler containing only MSCs is significantly lower compared to previous studies.

In our study, the efficacy of the MSC-only filler was lower than reported in some previous studies. One reason for this may be the use of a gastrointestinal fistula model, which more closely mimics clinical enterocutaneous fistula conditions and presents a harsher environment for filler engraftment.

We believe that MSCs alone were not sufficient to promote stable engraftment in this ischemic and inflammatory setting. In contrast, the combination of MSCs and HUVECs promoted angiogenesis, which likely contributed to improved filler survival and integration. This explanation has been added to the Discussion section (page 14, lines 15–20).

10. English and Organization:

Comment: This paper would benefit from English language improvement and a more coherent content organization.

Response: In response, we have thoroughly revised the manuscript to improve clarity, grammar, and overall readability. Particular attention was paid to refining the English language and ensuring logical flow across sections. We believe these improvements have enhanced the manuscript’s coherence and accessibility. If further corrections are required, we would be happy to address them.

Editorial Office Comments:

1. Formatting and Style Requirements:

Response: We have revised the manuscript to comply with PLOS ONE’s formatting and file naming guidelines.

2. Funding Information Mismatch:

Response: We have corrected and confirmed that the funding information is consistent throughout the manuscript.

3. Following financial disclosure

Response: We also added the following statement: “The funders had no role in study design, data collection and analysis, decision to publish, or preparation of the manuscript.”

4. Full Ethics Statement:

Response: A full ethics statement, including the name of the ethics committee and the approval number, has been added to the Methods section.

5. Data Availability Statement:

Response: We confirm that all data required to replicate the results are included in the manuscript or supporting information files. If needed, raw data files have been uploaded as Supporting Information.

6. Abstract Citations:

Response: All citations have been removed from the Abstract to comply with PLOS ONE’s guidelines.

We hope the revised manuscript and our responses fully address the concerns raised. We are grateful to the reviewers and the editorial team for their valuable feedback.

Sincerely,

Soichiro Hirasawa, on behalf of all authors

---

## [Decision Letter · Decision Letter 1]

4 Aug 2025

MSC and HUVEC co-cultured fillers overcome intractable fistula in a new mouse model

PONE-D-25-19698R1

Dear Dr. Hirasawa,

We’re pleased to inform you that your manuscript has been judged scientifically suitable for publication and will be formally accepted for publication once it meets all outstanding technical requirements.

Kind regards,

Keykavoos Gholami

Academic Editor

PLOS ONE

Additional Editor Comments (optional):

Reviewers' comments:

Reviewer's Responses to Questions

**Comments to the Author**

Reviewer #1: All comments have been addressed

Reviewer #2: All comments have been addressed

2. Is the manuscript technically sound, and do the data support the conclusions?

Reviewer #1: Yes

Reviewer #2: Yes

3. Has the statistical analysis been performed appropriately and rigorously?

Reviewer #1: Yes

Reviewer #2: Yes

4. Have the authors made all data underlying the findings in their manuscript fully available?

Reviewer #1: Yes

Reviewer #2: Yes

5. Is the manuscript presented in an intelligible fashion and written in standard English?

Reviewer #1: Yes

Reviewer #2: Yes

Reviewer #1: all comments have been addressed carefully. I think this manuscript is suitable for publication in PLOS ONE.

Reviewer #2: (No Response)

**Do you want your identity to be public for this peer review?** For information about this choice, including consent withdrawal, please see our Privacy Policy

Reviewer #1: No

Reviewer #2: No

---

## [Editor Report · Acceptance letter]

PONE-D-25-19698R1

PLOS ONE

Dear Dr. Hirasawa,

I'm pleased to inform you that your manuscript has been deemed suitable for publication in PLOS ONE. Congratulations! Your manuscript is now being handed over to our production team.

Kind regards,

on behalf of

Dr. Keykavoos Gholami

Academic Editor

PLOS ONE